# Visualizing defect dynamics by assembling the colloidal graphene lattice

Piet J. M. Swinkels [1], Zhe Gong[2], Stefano Sacanna [2], Eva G. Noya [3] & Peter Schall [1] ✉

Graphene has been under intense scientific interest because of its remarkable optical, mechanical and electronic properties. Its honeycomb structure makes it an archetypical two-dimensional material exhibiting a photonic and phononic band gap with topologically protected states. Here, we assemble colloidal graphene, the analogue of atomic graphene using pseudo-trivalent patchy particles, allowing particle-scale insight into crystal growth and defect dynamics. We directly observe the formation and healing of common defects, like grain boundaries and vacancies using confocal microscopy. We identify a pentagonal defect motif that is kinetically favoured in the early stages of growth, and acts as seed for more extended defects in the later stages. We determine the conformational energy of the crystal from the bond saturation and bond angle distortions, and follow its evolution through the energy landscape upon defect rearrangement and healing. These direct observations reveal that the origins of the most common defects lie in the early stages of graphene assembly, where pentagons are kinetically favoured over the equilibrium hexagons of the honeycomb lattice, subsequently stabilized during further growth. Our results open the door to the assembly of complex 2D colloidal materials and investigation of their dynamical, mechanical and optical properties.

Two-dimensional materials have attracted intense scientific interest, both from an application and a fundamental point of view, offering applications from light-weight materials to optoelectronic devices. These materials combine extraordinary mechanical, optical and electronic properties compared to bulk materials[1–3]. The most prominent representative, graphene, consists of a monolayer of carbon atoms bonded in a honeycomb lattice. The strong covalent bonds within the honeycomb lattice make the material particularly strong and light, while the honeycomb structure gives rise to a photonic and phononic band gap[4,5]. On a larger length scale, micrometre-size particles assembled into colloidal graphene, the analogue of atomic graphene assembled from colloidal particles, would open the door to two-dimensional multifunctional materials with photonic and phononic band gap for applications as 2D photonic and phononic crystals.

However, producing large defect-free single-crystalline graphene of both atoms and colloids remains a great challenge, crucially limiting its applications.

Structural defects are known to be central to all of graphene's properties, enabling among others band-gap tuning in graphene-based electronic devices[6]. However, while defects are introduced unavoidably during growth or added on purpose to tune mechanical and electronic properties, a comprehensive understanding of their formation is missing: because the trivalent carbon atoms or particles can arrange into a variety of polygons and structures, a coherent lattice exists even with defects, and rearrangements can take many paths. Despite advances in direct visualization of graphene defects using electron microscopy[6–8], defect kinetics and healing remain poorly understood, and defect-free graphene challenging to produce.

[1]Institute of Physics, University of Amsterdam, Amsterdam, the Netherlands. [2]Molecular Design Institute, Department of Chemistry, New York University, New York, NY, USA. [3]Instituto de Química Física Rocasolano, CSIC, Madrid, Spain. ✉e-mail: P.Schall@uva.nl

Although colloidal particles are several orders of magnitude larger than atoms, their phase behaviour and dynamics are governed by the same thermodynamics principles. Phase behaviour of both atoms and colloids is largely governed by thermal forces, which means we can use colloidal aggregation and crystallization as a simple model for atomic crystallization[9–11]. One advantage of colloidal systems is that defect formation[12–14] and dynamics[15,16] can be conveniently studied directly in real time with single particle resolution using optical microscopy[17,18], which remains challenging in atomic systems, especially under the harsh high-temperature atomic deposition used for graphene growth[19]. The recently-gained ability to synthesize anisotropic particles[20–23], in particular colloidal particles with attractive patches providing specific valency and bond angles, has opened a design space for assembling more complex structures such as molecule analogues[24–26] and covalently bonded crystals[27,28].

Simulations and experiment have shown that these colloidal molecules can grow into larger assemblies, yielding rich structures ranging from the kagome lattice to buckyball-like clusters[27,29]. Experimentally realizing these structures remains challenging, as they require fine control over specifically coordinated interactions, or purposeful geometric design to block kinetically favoured nonequilibrium routes, as recently shown for the realization of colloidal diamond[30,31]. In contrast to tetrahedrally coordinated diamond, graphene relies on the trivalent coordination of particles. Patchy particles with patches at 120° angles can mimic these covalent bonds; yet, achieving such valency and controlling these directed bonds on the

scale of $k_BT$, the thermal energy, remains challenging, but would open up the assembly of structurally complex 2D materials, and investigation of their structural and mechanical properties.

Here, we assemble colloidal graphene and elucidate the kinetic pathways of crystallization and defect formation of this 2D material. We form the graphene lattice using pseudo-trivalent patchy particles adsorbed at a substrate, and directly follow the crystallization, defect formation and healing with great temporal and spatial resolution using confocal microscopy. Fine control of the patch-patch bond strength allows observation of near-equilibrium assembly in analogy to high-temperature deposition of atomic graphene. From the number of saturated bonds and the bond strain, we determine the configurational energy of the lattice and follow its evolution during lattice rearrangement and healing. Our results reveal that the most prominent defect motif of colloidal and atomic graphene, a pentagon, is kinetically favoured in the early stages of graphene growth, and acts as seed for extended defects during subsequent growth. These results hint at the importance of the early stages of assembly in generating defect-free graphene.

## Experiments

We fabricate patchy particles from polystyrene (PS) and 3-(trimethoxysilyl)propyl methacrylate (TPM) spheres by colloidal fusion[23]. The synthesis yields tetrahedrally coordinated particles with a PS bulk and fluorescently labelled TPM patches (Fig. 1a, b). The particles have a diameter of $d = 2.0\,\mu m$ (see Supplementary Note 1) and a patch

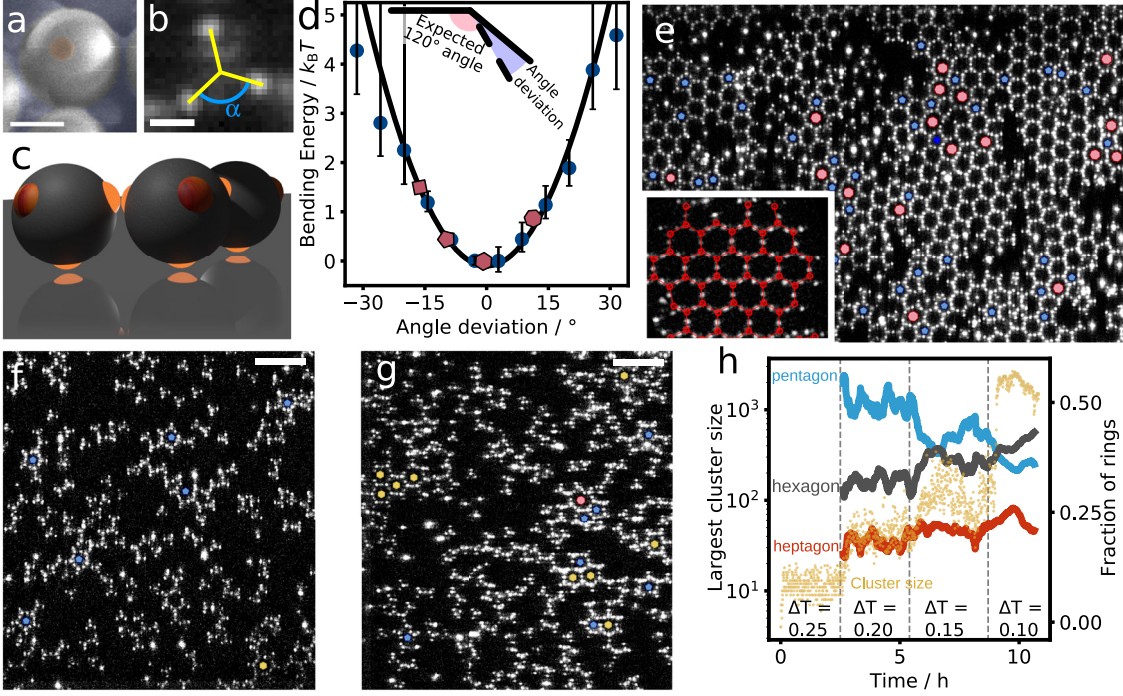

**Fig. 1 | Colloidal graphene flakes formed from trivalently coordinated particles.**
**a** Scanning electron microscopy (SEM) image of patchy particle, with patch highlighted in orange. **b** Confocal microscope image of patchy particle, highlighting the fluorescently labelled patches (bright spots). The projected angle between the patches is $\alpha = 120°$. Scale bars in (**a**) and (**b**) indicate 1 μm. **c** Reconstruction of the surface-bound tetramer patchy particles. One patch is attached to the surface, the remaining three patches available for bonding, making them effectively trivalent. The bonds are tilted slightly out of plane, hence deviating slightly from the ideal in-plane $sp^2$-like bonding. **d** Bending potential determined from angle fluctuations of three bonded particles (blue dots, see Supplementary Note 6 for details). Error bars indicate standard deviation. The black solid line indicates the harmonic fit $U_{bend} = \frac{1}{2}k_{bend}\theta^2$ with force constant $k_{bend} = 0.012\,k_BT/(°)^2$. The red $n$-gons show the resulting average bending energy per particle in rings composed of 4, 5, 6, and 7

particles (from left to right). **e** Confocal microscope image of honeycomb lattice of the pseudo-trivalent particles at $\Delta T = 0.05°C$ after 24 h of equilibration. Blue and red symbols indicate pentagon and heptagon defects. Inset: enlarged section with particle centres highlighted in red, showing the honeycomb lattice built out of 6-membered rings. **f, g** Confocal microscope images of the pseudo-trivalent particles at $\Delta T = 0.20°C$ (**f**) and $\Delta T = 0.15°C$ (**g**). With increasing attractive strength, particles assemble into small clusters preferentially containing pentagons (**f**), which grow into larger structures, favouring hexagons (shown in yellow) (**g**). Scale bars indicate 10 μm. **h** Fraction of pentagons (blue), hexagons (black), and heptagons (red), and largest cluster size (yellow scatter) as a function of time. Dashed lines delineate temperature changes. Hexagons grow at the expense of the pentagons. Source data are provided as a Source Data file.Source Data.

diameter $d_p \sim 0.2 \, \mu m$, sufficiently small to allow only single patches to bind with each other. To induce an effective patch-patch attraction of controllable magnitude, we suspend the particles in a binary solvent close to its critical point. The confinement of solvent fluctuations between the particle surfaces then causes attractive critical Casimir interactions on the order of the thermal energy, $k_B T$, tunable by the temperature offset $\Delta T$ from the solvent critical point, $T_c$[32–35].

We use a binary mixture of lutidine and water with lutidine volume fraction $c_L = 0.25$ close to the critical volume fraction $c_{L,c} = 0.27$[36], and solvent demixing temperature $T_{cx} = 33.95 \, °C$, and add 1 millimolar of MgSO$_4$ to screen the particles' electrostatic repulsion and enhance the lutidine adsorption of the hydrophobic patches[25]. The suspension is injected into a glass capillary with hydrophobically treated walls (see Supplementary Method 1) to which the particles become adsorbed via one of their patches at $\Delta T \leq 0.6 \, °C$. The resulting pseudo-trivalent particles diffuse freely along the surface (see Supplementary Note 2), until at $\Delta T \leq 0.25 °C$, the free patches start attracting each other, as illustrated in Fig. 1c. To observe near-equilibrium assembly, we slowly approach $T_c$ in steps of 0.05 °C starting from $\Delta T = 0.25 \, °C$, leaving the sample to equilibrate for 4 h at each step. The resulting slowly increasing patch-patch attraction ensures a near-equilibrium route to crystallization and mimics the slow cooling during high-temperature atomic deposition. The final binding energy is $12–15 k_B T$, as shown in Supplementary Note 6. We also measure the bending potential from bond angle fluctuations of two particles bonded to a central one (see Supplementary Note 6 for details). The resulting bending potential is closely harmonic as shown in Fig. 1d, allowing us to determine the bending force constant from the parabolic fit. We follow the structure and defect formation processes at the particle scale using rapid bright-field and confocal microscope imaging to track both the particles' centre of mass and fluorescent patches to determine the bond angles with their neighbours (see Methods and Supplementary Note 3).

## Results

The final assembled structure shows large flakes of honeycomb lattice, as shown in Fig. 1e. In the lattice, each particle has 3 bonds, at 120° angle with respect to each other, resulting in the repeating 6-membered hexagonal ring motif characteristic of the honeycomb lattice of graphene, as clearly shown in the inset. Indeed, the observation of the honeycomb lattice at our colloidal particle densities is in agreement with simulations of surface-confined trivalent particles predicting the honeycomb lattice for intermediate particle densities[37]. Besides the hexagonal honeycomb motif, however, we notice the presence of 5- and 7-membered rings, pentagons and heptagons, often sitting at boundaries of the honeycomb flakes. The crystal flakes and defects remind of those of atomic graphene grown by chemical or physical vapour deposition. Furthermore, we observe the formation of an amorphous layer when we quench the trivalent particles to high interaction strength (see Supplementary Note 4), in line with the amorphous structures observed in low-temperature vapour deposition[38].

To obtain further insight into the growth of the colloidal honeycomb lattice, we follow the initial stages of assembly at low interaction strength. Surprisingly, many 5-membered rings form initially, as shown in Fig. 1f, where small, open pentagon clusters are prevalent (blue dots). As the interaction strength increases, particle clusters grow, and more hexagon motifs, accompanied by heptagon motifs are observed (Fig. 1g, yellow and red dots, respectively). The dynamic evolution of the different motifs is clearly shown in Fig. 1h, where we plot the fraction of pentagons, hexagons, and heptagons, together with the largest cluster size as a function of time (see Supplementary Note 5 for more details). Initially, pentagons are the clear majority, while with increasing attraction, as larger clusters form, the number of hexagons grows at the expense of pentagons until they become the majority and we observe the fully grown flakes in Fig. 1e.

## Defects: grain boundaries and vacancies

The fully grown flakes show characteristic defects; most prominent examples, grain boundaries and vacancies, are shown in Fig. 2. The grain boundary consists of a line of alternating pentagons and heptagons bounding crystalline regions with different orientation above and below, indicated by the green dotted line in Fig. 2a. The scar of pentagons and heptagons causes a distinct shift in the orientation of the crystal: the honeycomb grains are rotated by approximately 17°. These grain boundaries are commonly observed in colloidal graphene: the combination of 5- and 7-membered motifs makes them geometrically most compatible with the honeycomb lattice, as shown schematically in Fig. 2b. In the ideal lattice, all bond angles are 120°; in contrast, pentagons exhibit internal angles of 108°, incompatible with the honeycomb lattice, making adjacent 6-membered rings unfavourable. Instead, the system typically forms the more favourable combination of alternating pentagons and heptagons, cancelling most of the angular mismatch, see Fig. 2b. Hence, the presence of a pentagon facilitates neighbouring heptagons, which in turn promote neighbouring pentagons, stabilizing the grain boundary. This is confirmed when looking at the bond bending energy (Fig. 1d): starting from a pentagon, growing two adjacent hexagons at the final temperature $\Delta T = 0.05 °C$ imposes a bending energy cost of $\sim 0.43 k_B T$, corresponding to an angle deviation of 12° while growing an adjacent heptagon and hexagon imposes a bending energy of only $\sim 0.14 k_B T$, corresponding to an angle deviation of 3.4°, three times less. Similar grain boundaries of alternating pentagons and heptagons are found in atomic graphene[39,40]. Although the details of the inter-atomic attractions are different from those of the colloidal particles, the same geometric argument underlying the typical pentagonal and heptagonal motifs applies.

To obtain insight into the bond bending strain and defect energy, we plot the bending energy as a function of distance away from the interface (in terms of number of bonds) in Fig. 2c. Close to the grain boundary, there is an increased bending energy cost despite the energetic advantage of pentagon-heptagon combinations, maximizing at $\sim 0.4 k_B T$ per particle. The large variance of the data points reflect the thermal energy, which is of the same order as the defect energy cost. The latter quickly drops away from the interface, being virtually absent after just one layer of hexagons. Vacancies indicate defects where one or more particles are missing in the honeycomb lattice. An example of a divacancy, where two particles are missing, is shown in Fig. 2d, while a bigger poly-vacancy is shown in Fig. 2e. In the first case, the surrounding honeycomb lattice is not much perturbed: the crystal structure remains intact, and the orientation of the 6-membered rings does not change. In the second case, the vacancy has a large effect on the surrounding lattice; the lattice is deformed and partly collapsed upon itself.

Vacancies are of interest in atomic graphene, as they can unlock desirable material properties, like catalytic activity and improved electronic properties[41–43]. However, CVD-grown atomic graphene does not normally show vacancy defects, even though experiments show that vacancies are generated during the early stages of the CVD process[44]. Unlike the substrate-adsorbed colloidal particles, carbon atoms can approach vacancies from outside the plane during CVD growth, filling the vacancies with feedstock carbon[44]. Hence, unlike vacancies in colloidal graphene, vacancies in atomic graphene anneal in the CVD process, and irradiation or chemical treatment is used to induce them. These generated vacancies typically reconfigure to a (slightly) lower-energy structure that contains fewer dangling bonds. For instance, a divacancy can reconfigure into two 5-membered and one 8-membered ring[45]. In contrast, the divacancy in Fig. 2d is stable and does not reconfigure. We associate this with the more rigid bonds of the short-range critical Casimir interaction potential[46], making reconfigurations in colloidal graphene unlikely due to high energy barrier (see Supplementary Note 7), unlike their atomic counterpart[7,47,48].

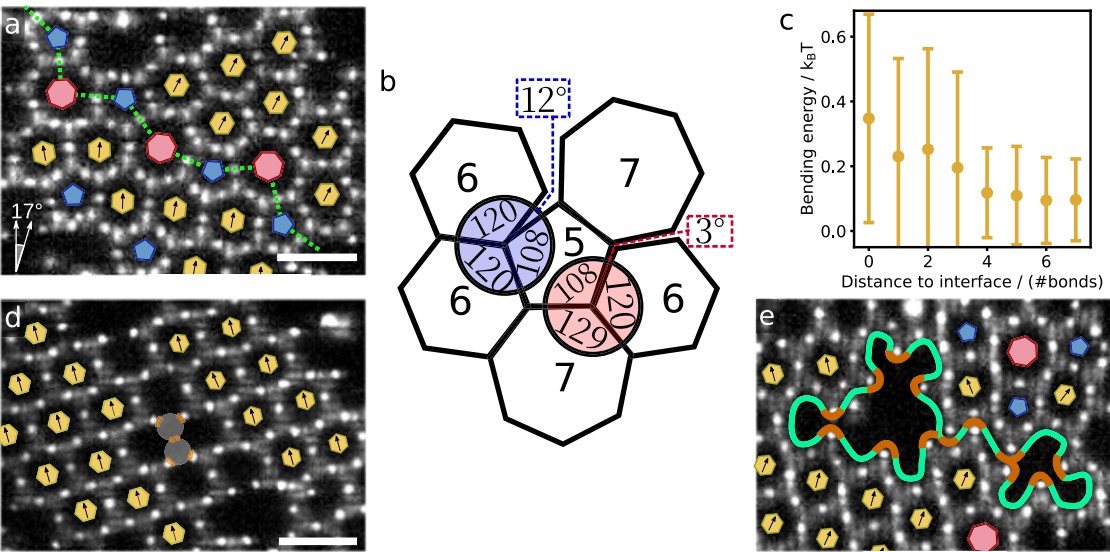

**Fig. 2 | Defects in colloidal graphene. a** Confocal microscope image of a typical grain boundary consisting of pentagons and heptagons. Green dotted boundary separates grains rotated by $\theta$ - 17° with respect to each other (left lower inset), which matches observations in atomic graphene[64]. **b** Schematic indicating the compatibility of 5-, 6- and 7-membered rings. Two 6-membered rings and a 5-membered ring leave a 12° angular mismatch, while a combination of one 5-, one 6-, and one 7-membered ring leaves a mismatch of only 3.4°. **c** Bond bending energy as a function of distance (in terms of the number of bonds) away from the boundary in **a**. Variances indicate the standard deviation of the energy distribution, which is mostly due to thermal noise. **d** Confocal microscope image of a divacancy in colloidal graphene. The missing patchy particles are illustrated in grey with orange patches. **e** Confocal microscope image of a multivacancy. The boundary of the vacancy is indicated with a green line, and dangling bonds with orange line segments. In all panels, 6-membered rings are indicated with yellow hexagons, 5-membered rings with blue pentagons, and 7-membered rings with red heptagons. The orientation of hexagons is indicated with a black arrow. All images are taken at $\Delta T = 0.05$ °C. Scale bars indicate 5$\mu$m. Source data are provided as a Source Data file.Source Data.

## Defect formation

To obtain further insight into the origin of colloidal graphene defects, we follow the defect formation process more closely. Pentagons, generated early in the assembly process, can act as nucleation sites for grain boundaries, as shown in Fig. 3a–c. The grain boundary grows from a pre-existing pentagon that forms at $\Delta T = 0.10$ °C (blue arrow in Fig. 3a, b), and subsequently leads to the formation of a heptagon, as shown in Fig. 3b, c (pink arrow). The heptagon again promotes the formation of a pentagon (orange arrow), so that a grain boundary of successive alternating pentagons and heptagons is established after 7 min, as shown in Fig. 3c. This trend is reflected in the growth of heptagons in Fig. 1h: pentagons can convert into hexagons, or promote the growth of neighbouring heptagons, which stabilizes the structure as a whole. This process is facilitated by the initial prevalence of pentagons. Yet, this prevalence is surprising, as it is energetically more favourable to form the equilibrium hexagonal motif. To obtain more insight, we look closely at the formation of individual rings in the early stages. Energy traces for the formation of isolated pentagons and hexagons are plotted in Fig. 3d, while the corresponding cluster conformations are illustrated in Fig. 3e. The formation of the pentagon (blue data and line) starts from initially three particles, which after addition of the fourth and fifth particle immediately close into a 5-particle ring before a sixth particle arrives. Even subsequent opening and closing events due to thermal fluctuations, visible as the three subsequent peaks, are not successful to incorporate a sixth particle (see Supplementary Note 8 for more details). In the case of the hexagon (yellow data and line), the ring remains open after addition of the fifth particle and is able to accumulate a sixth particle before closing, resulting in a hexagon with a lower energy. We, therefore, conclude that the formation of pentagons is kinetically favoured. An analogous kinetically favoured pathway is observed in the assembly of colloidal diamond from tetrahedral particles: 5-membered motifs are kinetically favoured, hindering the formation of the diamond lattice and making it difficult to assemble colloidal diamond[25,29,30]. Similarly, our results on

colloidal graphene demonstrate that kinetically favoured pentagons hamper the formation of the equilibrium hexagonal motif, leading to grain boundaries in later stages that limit the growth of the honeycomb lattice. An equivalent mechanism may be effective in atomic graphene; while the role of pentagons in the formation of grain boundaries has not yet been reported, observations of pentagon formation in early growth of atomic graphene support this possibility[49,50].

Grain boundaries can also emerge from the merging of crystal grains as shown in Fig. 3f–h. The system minimizes the number of dangling bonds by stitching the two crystals together, while reconfigurations of the misaligned grains lead to a scar of pentagons and heptagons after 120 min (see Fig. 3h and Supplementary movie 1). Grains can also merge seamlessly, see Supplementary Note 9.

To form a single crystal of atomic graphene, the single seed approach, nucleating and growing graphene from a single point, has been applied with success[51]. This single-seed approach is in principle applicable to our colloidal system owing to the temperature sensitivity of the critical Casimir interaction, yet may be difficult to achieve in practice due to the small temperature window. For colloidal systems, it has been suggested that the formation of pentagons could be suppressed by introducing two types A and B of patchy particles that only bind to the other type but not to themselves, forcing even-numbered rings[52]. While this is an elegant method, application to our critical Casimir system is not straight forward as interactions cannot easily be specifically encoded. We discuss these and more strategies and their suitability for the critical Casimir system in Supplementary Note 10.

## Defect evolution

To elucidate the slow reconfiguration of graphene defects in more detail, we follow the graphene polycrystal over a time interval of 9 h. Snapshots of the initial configuration, and after 4.5 and 9 h are shown in Fig. 4a–c. The red and yellow delineated regions show examples of static and highly dynamic grain boundaries, respectively. The former shows no reorganization: any translation or rotation matches the

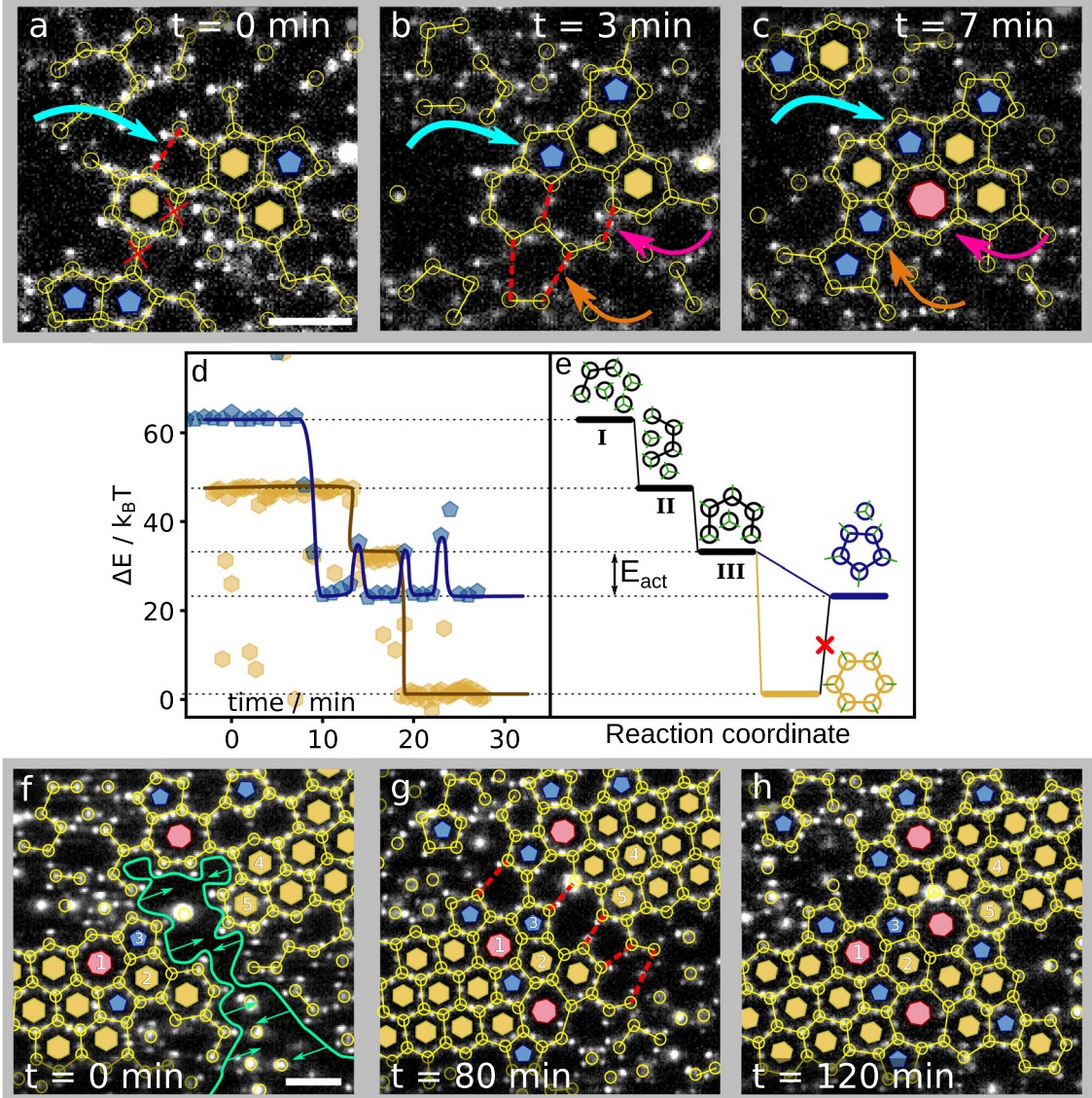

**Fig. 3 | Origin of defects of colloidal graphene. a–c** Annotated confocal microscope images show the spontaneous formation of a grain boundary at $\Delta T = 0.10\,°C$, where the honeycomb crystals are still small. In (**a**), two particles are about to bond at the red dotted line (blue arrow), forming a 5-membered ring. In (**b**), the 5-membered ring has formed, and promotes a neighbouring 7-membered ring (pink arrow). The 7-membered ring in turn promotes a 5-membered ring next to it (orange arrow). In (**c**), we show the final structure that has grown spontaneously. **d, e** Energy-time traces of an isolated pentagon (blue) and hexagon (yellow) forming from three and four particles, respectively. The former closes immediately into a 5-ring after addition of two particles and keeps trying to incorporate a sixth particle by repeated opening, while the latter remains in the open state until a sixth particle attaches. **f–h** Annotated confocal microscope images show the formation of a grain boundary through the merging of two grains at $\Delta T = 0.05\,°C$. Two larger grains approach each other, indicated with green lines and arrows in (**f**). In (**g**), the grains have drifted closer together and red dashed lines indicate where particle bonds will form. In (**h**) the two grains have merged into one, and alternating 5- and 7-membered rings form their boundary. The rings numbered 1 to 5 are the same in each frame. Scale bars in (**a**) and (**f**) indicate 5 µm. Source data are provided as a Source Data file.Source Data.

movement of the entire crystal and the grain boundary is completely frozen. In contrast, the yellow delineated region close to the junction of multiple grains shows significant reconfiguration. The initial monovacancy, divacancy, and larger polyvacancy (Fig. 4a) merge into a bigger polyvacancy after $t = 4.5\,h$ (Fig. 4b), which upon further reconfiguration evolves into a monovacancy, divacancy, and a bigger polyvacancy after $t = 9\,h$ (Fig. 4c, see Supplementary Movie 2 for the full process). The enhanced dynamics is confirmed in a more detailed particle-scale analysis as shown in Supplementary Note 11. To elucidate the underlying driving force, we follow the total bond energy of the lattice over time. We include two energy contributions: energy costs due to unsaturated bonds, and energy costs due to structural distortions, where we consider only contributions from bond bending. The

resulting total energy as a function of time (Fig. 4d) reveals an energy landscape with maxima and minima, which clearly decreases for the yellow region, while it remains fairly constant for the red region. The data suggests that the yellow region slowly transitions towards a more favourable lower-energy state, while moving through the energy landscape, unlike the red region that cannot easily lower its energy.

To further elucidate the reconfiguration process, we plot the two energy contributions separately in Fig. 4e. Both are of similar order of magnitude, while the dangling bond contribution dominates the total energy drop. Interestingly, bending and dangling-bond contributions show opposite behaviour in the first case, revealing the system's frustration (closing of bonds leads to lattice distortions and vice versa), while in the dynamic case, the two contributions decrease in parallel,

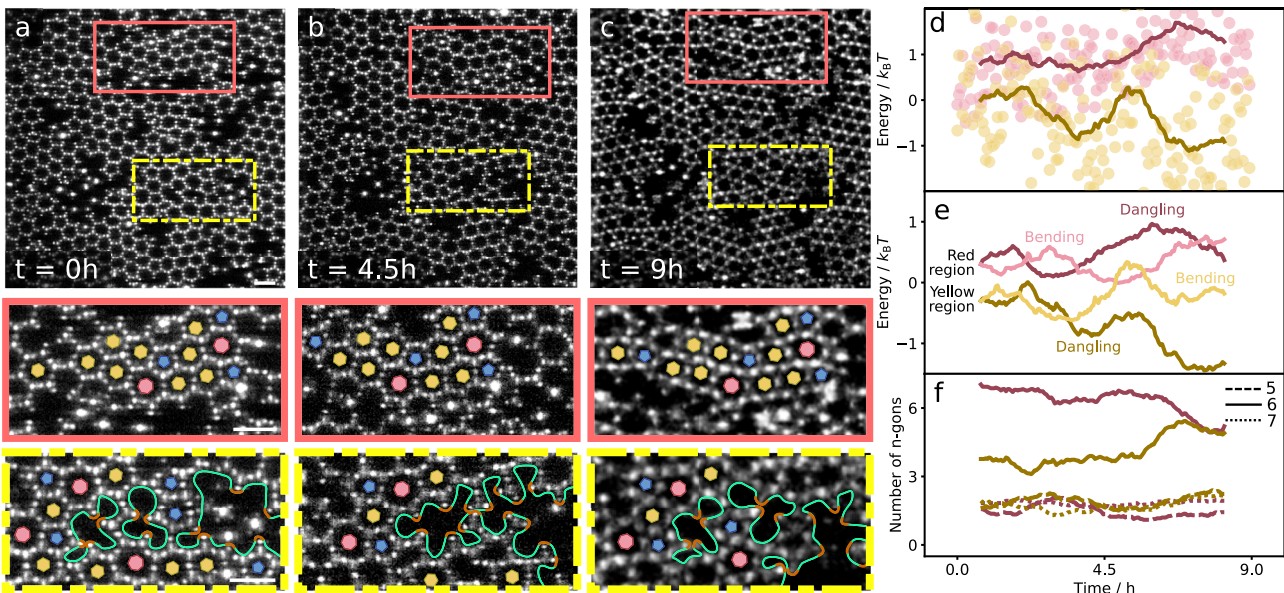

**Fig. 4 | Defect evolution in colloidal graphene. a–c** Top: Confocal microscope images showing a colloidal graphene polycrystal in its initial configuration (**a**) and after 4.5 (**b**) and 9 h (**c**) of equilibration at $\Delta T = 0.05\,°C$. Red and yellow rectangles demarcate regions with static and highly dynamic grain boundaries, respectively. Bottom: Enlarged sections corresponding to the demarcated regions on top. In these sections, 6-membered rings are indicated with yellow hexagons, 5-membered rings with blue pentagons, and 7-membered rings with red heptagons. The green solid lines bound poly-vacancies, the orange line segments indicate dangling bonds. Vacancies with 3 dangling bonds are typically monovacancies, vacancies with 4 dangling bonds divacancies. Scale bars in (**a**) correspond to $5\,\mu m$. **d** Total energy per particle computed from dangling bonds and bond bending as a function of time. Red and yellow dots and lines correspond to red and yellow enclosed regions. Dots represent individual frames, solid lines are running averages over 1 h (30 datapoints). The energy curves are shifted with respect to each other for clarity. **e** Energy contributions from dangling bonds and bond bending to the total energy plotted in (**d**), running averages of 1 h. Dark colours indicate dangling bond, and light colours bending contributions. **f** Number of pentagons (dashed lines), hexagons (solid lines), and heptagons (dotted lines) present in the two sections as a function of time. The increasing number of hexagons reflecting the closing of these motifs is in line with the decreasing dangling-bond energy in **e**. Source data are provided as a Source Data file.Source Data.

leading to energetically more favourable configurations. Apparently, the interplay of lattice distortion and dangling bond saturation determines the reconfiguration process. This is further corroborated in Fig. 4f, where we show the evolution of the number of hexagons (solid lines), pentagons and heptagons (dashed and dotted lines). The increasing number of hexagons accompanying the decreasing bending and dangling bond energy highlights the system's approach to the energetically most favourable honeycomb lattice, in contrast to the static case. The number of pentagons and heptagons changes only slightly in both cases. Our time and particle-resolved observations of colloidal graphene allow detailed insight into the defect dynamics of this important 2D material, highlighting the strong dynamical nature of large vacancies and the corresponding changes in the energy landscape.

It is interesting to compare the assembly in our system with that of curved 2D systems. Colloids can crystallize into hexagons on a flat surface, but not a curved one. However, introducing heptagons and pentagons to the lattice makes it possible to tile a curved surface (think of the pentagons on a football or on a viral shell)[53,54]. On such surfaces, pentagons and heptagons are actually part of the minimum-energy structure[53,55,56]. This is an interesting contrast to our planar lattice, for which penta- and heptagons are defects, and not part of the equilibrium structure.

## Discussion

Trivalent colloidal particles adsorbed at a substrate form the colloidal analogue of graphene, allowing direct observation of its crystallization and defect dynamics. The fine interaction control opens near-equilibrium crystallization and annealing pathways. We find that colloidal graphene defects originate in the early stages of crystallization from pentagonal motifs that are kinetically favoured over the equilibrium hexagonal motifs, further stabilized by adjacent heptagonal motifs, together forming stable grain boundaries. These results are reminiscent of high-resolution electron microscopy observations of atomic graphene, which however are limited to fully grown graphene, and cannot access the initial stages of crystallization. Grain boundaries and extended polyvacancies reconfigure towards lower-energy states by an interplay of dangling-bond saturation and lattice distortions, ultimately increasing the number of hexagons. Strategies for synthesizing large atomic graphene crystals tend to circumvent this issue by either generating as few graphene seeds as possible[51], or aligning graphene islands before merging[57]. While crystallization and defect dynamics in atomic graphene are ultimately governed by quantum mechanics, we expect that in the high-temperature limit studied here, where the quantum-mechanical states become quasi-continuous, our colloidal system provides a good model. Yet, differences arise from the different form of the potential and the different nature of the in-plane (covalent) versus interlayer bonding (van der Waals interaction). Despite these differences, the energy scales are similar: at typical high-temperature deposition (~800 K) the covalent bond energy of graphene on a metal substrate is ~1 eV[58], corresponding to $14.5\,k_BT$ - similar to our critical Casimir bond strength of $12-15\,k_BT$. During higher temperature annealing procedures of graphene (~2000 K) the corresponding energy ($6\,k_BT$) is lower, allowing together with the higher atomic attempt frequency for faster dynamics, with rearrangements occurring on the order of seconds[59] rather than hours as in our colloidal case. The assembly of colloidal graphene demonstrates the increasing control over bottom-up assembly of complex materials. The honeycomb lattice is of specific interest because it is the simplest metamaterial exhibiting a photonic and phononic bandgap[4,5,60,61], and topologically protected states. While achieving the structural complexity of macroscopic mechanical metamaterials remains still a challenge, our results demonstrate that the necessary structural

motifs can be assembled using patchy particles, opening the door to microscale mechanical metamaterials[4].

## Methods

### Sample details and critical Casimir interactions
Tetrapatch particles (diameter of 2.0 μm, patch diameter of approximately 0.2 μm) are dispersed in a binary solvent of 25% 2,6-lutidine (≥99.0%, Sigma Aldrich) and 75% milliQ water with 1 millimolar $MgSO_4$ (≥99.5%, Sigma-Aldrich). The particles are washed several times in the water-lutidine mixture. The resulting particle dispersion is injected into a silanized hydrophobic glass capillary (Vitrotubes, Rectangle Boro Tubing 0.20 × 2.00 millimetres) and sealed with teflon grease (Krytox GPL-205). The full silanization procedure is given in Supplementary Method 1.

### Confocal microscopy and particle tracking
Particles are left to sediment to the bottom of the sample at room temperature before measurements. We tilt the sample slightly during sedimentation, so a small density gradient is present in the sample. We then heat the sample to approximately 33.35 °C ($\Delta T \approx 0.6$ °C), which causes one of the particle patches to attach to the sample wall. For heating, we use a well-controlled temperature stage in combination with an objective heating element to obtain a relative temperature accuracy of about 0.01 °C with minimal temperature gradients.

In an experiment, we typically heat a sample to a certain $\Delta T$ below the phase separation temperature of the water-lutidine mixture, inducing critical Casimir attraction between patches. The structures then grow by two-dimensional diffusion in the plane. No mixing is necessary. We investigate the structures as they form using a 100x oil-immersion objective.

We image the assembled structures using confocal microscope image stacks, sometimes alternating with bright field images. The images are processed using particle tracking software (Trackpy[62]) to determine the centre of each patchy particle. Because one of the patches is attached to the glass, each particle centre is approximately given by this lower patch. The three remaining patches are suitable for bonding, and form a single blob with another bonded patch if bonded. This tells us if two nearby particles are actually bonded or merely randomly meeting. For more tracking details, see Supplementary Note 3.

### Radial and Bending potential of a patchy particle
The critical Casimir potential is short ranged, its magnitude and range are set by the temperature offset $\Delta T = T - T_c$ to the critical temperature, $T_c$. Further factors that determine the magnitude of the potential are the absorption preference of the surfaces (particle patches, glass surface), and the composition of the binary solvent, as detailed in ref. [46]. By formulating the critical Casimir potential model presented in ref. [46] for patchy particles and benchmarking it onto our patchy particles as shown in ref. [63], we determine the attractive minimum of the pair potential to be $12-15k_BT$ in the studied temperature range of $\Delta T = 0.25-0.1K$. The full potential is given in Supplementary Note 6.

The bond-bending potential is harder to estimate theoretically, but can be determined from experimental measurements by following clusters of three particles, and tracking the fluctuations of the internal angle of these particles. The resulting angle probability distribution can then be converted to a potential, as shown in Fig. 1d, using the Boltzmann distribution. The full procedure is given in detail in Supplementary Note 6.

## Data availability
The data that support the findings of this study are available from the corresponding author upon request. Source data of all plots are provided with this paper. Source data are provided with this paper.

## Code availability
The custom-made codes used to analyze the microscope images are available from the corresponding author upon request.

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

## Acknowledgements

P.S. acknowledges funding through Vici grant 680-47-615 from the Netherlands Organization for Scientific Research (NWO). S.S. acknowledges support from the NSF CAREER award DMR-1653465. E.G.N. acknowledges funding from Agencia Estatal de Investigación and Fondo Europeo de Desarrollo Regional (FEDER), Grant No PID2020-115722GB-C21.

## Author contributions

P.J.M.S. and P.S. conceived the study. P.J.M.S. performed the experiments and analysed the data with help of E.G.N. Z.G. and S.S. made the particles. P.J.M.S. and P.S. wrote the manuscript. All authors discussed the data and reviewed the manuscript.

## Competing interests

The authors declare no competing interests.
