## [Peer Review File · Nature Communications]

Visualizing defect dynamics by assembling the colloidal graphene latticeREVIEWER COMMENTS

Reviewer #1 (Remarks to the Author):

The authors study the assembly of patchy tetrahedrally coordinated particles adsorbed at a flat substrate. Using Confocal microscopy, the authors observe the growth of these particles into the two-dimensional colloidal graphene by slowly tuning the temperature. In particular, the authors monitor the assembly of defects including grain boundaries and vacancies as the lattice grows.

I think that the results of the paper are very interesting. The authors are able to directly monitor the crystallization and the kinetics of the formation of defects and "healing". Their work provides insight into the kinetic pathways of crystal growths and defect dynamics. Formation of the large defect-free single-crystalline graphene remains a great challenge and the study like the one given in this paper is a crucial step in the production of defect-free two-dimensional lattices.

I think that the paper is relatively well-written. However, the authors need to discuss more the literature and our current understanding of the formation of disclinations during crystallization and emphasize more how this work is different from the previous work. The papers will be more suitable for the publication if the authors consider the following items:

1. This is an experimental paper. The authors do not discuss their experimental approach neither in the abstract or in the introduction. The use of confocal microscopy and other techniques should be mentioned in the introduction. Is this a novel approach for defining the growth of defects during crystallization? The authors do not give enough background about the problem.

2. The authors state, "Our results reveal that the most prominent defect motif of colloidal and atomic graphene, a pentagon, is kinetically favoured in the early stages of graphene growth..". I am not sure what the authors mean by kinetically "favoured"? Do the authors think that pentagons are not equilibrium structures and thus they are kinetically trapped structures? If the attractive interaction is weak, then entropic forces might become considerable and the system might prefer the formation of pentagonal defects. Can they prove that the pentagons are not the result of minimization of the free energy at the weak interaction? Since the authors believe that pentagons are the kinetically trap structures, can they change the pathway?

3. How do the authors know that the attractive interaction between colloids is because of the Casimir forces? How about van der Waals forces? Can the authors explain how they calculate the magnitude of the Casimir forces in their system? Can the fluid deformation around the particles give a rise to an attractive force? Is there another system in which the casimir forces create a macroscopic lattice?

4. Have the authors tried different annealing pathways to overcome the energy barriers and to prevent the formation of defects when the attractive interaction is strong? If it is a kinetic trap, how universal their results are?

5. The authors mention, "Simulations and experiment have shown that these colloidal molecules can grow into larger assemblies, yielding rich structures ranging from the kagome lattice to buckyball- like clusters." However, they refer neither to the experimental nor to the theoretical papers. For example, the paper by Weitz group: Grain Boundary Scars and Spherical Crystallography published in Science in 2003 discuss in detail why scars exist in spherical crystals. Or Li et al. analytical results show how disclinations interact with each other in curved space (PNAS 115, 10971–10976 (2018)) as in the case of buckyball and thus explain the symmetry of spherical viruses. In the curved systems, the presence of scars and pentagonal defects lowers the energy. In the system that the authors use, the minimum energy structure is defect-free. The paper will become more interesting if the authors compare the origin of defects in flat and curved space.

6. In supplementary Note 5, the authors indicate that the particle cluster grows linearly as a chain. Is this also a kinetic effect? This system should be completely out of equilibrium assuming a configuration with a considerable high line tension.

7. The supplementary Note 6 is important for understanding the major energy components discussed in the main text. It is hard to understand the paper without the discussion of these energies in the main text. I recommend that the authors move some parts of Note 6 to the main texts. I think that the paper will be easier to follow if the authors showed the relevant equations and add some relevant theories to the paper.

Reviewer #2 (Remarks to the Author):

This manuscript by Swinkels et al., reports on the self-assembly of colloidal graphene - the colloidal analog of atomic graphene - from pseudo-trivalent patchy particles adsorbed on a substrate, and elucidates the kinetic pathways of crystallization and defect formation. In this case, controlled attractive Casimir interactions between the patches are induced to drive the self-assembly. The elucidation of the crystal growth with particle-level resolution reveals that pentagons are prevalent at the initial stage, but the number of hexagons grows at the expense of pentagons as the strength of the attractions is increased. The formation of alternating pentagons and heptagons is suggested to stabilize grain boundaries.

The work has been carried out thoroughly and the results are significant from the perspective of both fundamental understanding and potential applications. The methodology has been presented in sufficient detail. The manuscript is overall well presented.

Given the importance of atomic graphene as a multifunctional material, and the fundamental understanding about kinetic pathways of crystallization and defect formation that the present study on colloidal graphene lends, I recommend for the publication of the manuscript in Nature Communications after revision.

The authors are encouraged to consider the following questions / comments while revising the manuscript.

1. The authors note that the pentagons and heptagons appear in pairs, as apparent in Figure 3. However, it appears in Figure 1g that the number of hexagons grows at the expense of pentagons as the strength of the attractions is increased, while the number of heptagons also shows an overall growth, albeit slightly. Can the authors comment on this observation?
2. The x-axis label of Figure 2b requires further clarity. It is somewhat different from what is stated in the figure caption and in the text. Perhaps a pictorial illustration will help.
3. Given the importance of the analysis in terms of the bond bending energy, the authors should consider including a concise description of bond bending energy in the main body of the manuscript and state how it is calculated in the Methods section (rather than in the Supplementary Material) so that the manuscript can be read seamlessly.
4. In the concluding section, the authors argue that the energy scales are similar in atomic graphene and colloidal graphene in certain conditions. It is not clear then why the Stone-Wales defects, rather common in atomic graphene, are not seen in colloidal graphene. The comparison with atomic graphene with respect to the energy of formation of Stone-Wales defects should be elaborated.
5. A recent computational study [PNAS 118, e2109776118 (2021)] has highlighted the importance of the selection of six-member rings for the self-assembly of colloidal diamond. In this context, it will be worthwhile to comment on how the selection of six-member rings, in preference to five- and seven-member rings, along the crystallization pathways can possibly be encoded to improve the self-assembly kinetics as well as the quality of colloidal graphene thus formed.

Reviewer #3 (Remarks to the Author):

In "Visualizing defect dynamics by assembling colloidal graphene," Swinkels and co-authors fabricate tetra-patch colloidal spheres, assemble these into a monolayer within a binary solvent, and then use temperature to control the Casimir forces responsible both for patch-surface and patch-patch interactions. Upon initiating the attractive patch-patch interactions, they track the formation and growth of 'graphene-like' crystallites, along with a number of defect types and grain boundaries. The authors claim that the observation of defect dynamics in the colloidal graphene-analogue could aid in understanding some aspects of graphene synthesis.

To my knowledge, this is the first time the colloidal-fusion approach [Gong, Z., Hueckel, T., Yi, G.-R., & Sacanna, S. (2017). Patchy particles made by colloidal fusion. *Nature*, 550(7675), 234–238. <https://doi.org/10.1038/nature23901>] has been used to produce patchy colloids to study the dynamics of 2D particle assembly. I found the manuscript highly interesting, with an impressive methodology to realise and analyse the patchy colloidal polycrystals. From the results, highlights are the observation of predominantly pentagonal nuclei which evolved into grain boundaries, and the quantification of the bending energy. I think this work will be of interest to the patchy colloidal community, and also have much broader significance to the field of 2D crystallisation. I would like to recommend publication in *Nature Communications*, but I have key comments which first need to be addressed. Point 7 below is particularly significant.

1. 'Colloidal graphene' already has a different meaning in the graphene synthesis community, referring to a dispersion of graphene platelets [Li, L., & Yan, X. (2010). Colloidal Graphene Quantum Dots. *The Journal of Physical Chemistry Letters*, 1(17), 2572–2576. <https://doi.org/10.1021/jz100862f>]. Although this label makes sense in the main text, in the title it is misleading and could be passed over by the mesoscale assembly community.

2. In Figs. 1e and 1f, the hexagonal labels (grey hexagons) are very hard to distinguish from the greyscale images – could they be recoloured?

3. Fig. 2c is referenced before Fig. 2b. Consider reorganising Fig 2.

4. I find Fig. 2b and associated discussions confusing, and the explanation needs to be expanded. You say:

"To obtain insight into the bond bending strain and defect energy, we plot the bending energy as a function of distance away from the interface in Fig. 2b. Close to the grain boundary, there is an increased bending energy cost despite the energetic advantage of pentagon-heptagon combinations, maximizing at $\sim 0.5k_B T$ per particle. This energetic penalty quickly drops away from the interface, being virtually absent after just one layer of hexagons."

"Plot the bending energy as a function of distance away from the interface" – does this mean that: (1) you evaluate the average deviation in bond angles from 120° , and plot this against the distance away from the interface (2) you convert the bond angle deviation to bending energy, using supplementary Fig. 7b?

Furthermore, you say that you "Plot the bending energy as a function of distance away from the interface," but Fig. 2b is a plot of the bending energy vs. 'number of bonds removed from defect'. I don't understand what you mean by this plot, or how it related to the discussion on bending energy vs distance from the interface.

How did you construct the "bending energy as a function of distance away from the interface" – did you observe a large number of interfaces and produce averages (in which case did you only select interfaces containing 5-7 defect chains, or were other types included in your sample)? Alternatively, is this evaluated from a single image? Either way, I would expect to see error bars in Fig. 2b representing the standard deviation about this average at each distance.

5. Given the importance of the bending energy to the discussions in the main text, I find supplementary Figs. 7b and 7c highly important, and I feel they should be in the main text. The construction of supplementary Fig. 7b need to be explained more. Currently in the SI you say "By

following three bonded particles and tracking the fluctuations of their bond angles, we can determine the bending energy by assuming a Boltzmann distribution. The resulting bending energy as a function of angle, determined from the probability distribution of bond angles, is shown in Supplementary Fig. 7b." What was your sample size, and what are the uncertainties/error bars in Fig. 7b? Including more detail on this is important, e.g. showing the histogram and Boltzmann fit, as otherwise the reader cannot evaluate the reliability or accuracy of your bending energies.

6. In Fig. 3d, the pentagon energy trace is red and the hexagon energy trace is blue, whereas throughout the rest of the paper, pentagons are identified as blue points, hexagons as grey, and heptagons as red. Can you make the colouring consistent?

7. Fig. 4 doesn't appear to support your distinction between 'static and dynamic' grain boundaries. You claim that, "The red and yellow delineated regions show examples of static and highly dynamic grain boundaries, respectively. The former shows no reorganization: any translation or rotation matches the movement of the entire crystal and the grain boundary is completely frozen. In contrast, the yellow delineated region close to the junction of multiple grains shows significant reconfiguration." However, observing Figs. 4d-f, the rearrangement dynamics appear quite similar, (particularly Fig. 4f). To investigate your claim, I would expect to see an analysis similar to, for example, [Singh, N., Sood, A. K., & Ganapathy, R. (2020). Cooperatively rearranging regions change shape near the mode-coupling crossover for colloidal liquids on a sphere. *Nature Communications*, 11(1), 4967. <https://doi.org/10.1038/s41467-020-18760-7>] (Fig.3), or [van der Meer, B., Qi, W., Fokkink, R. G., van der Gucht, J., Dijkstra, M., & Sprakel, J. (2014). Highly cooperative stress relaxation in two-dimensional soft colloidal crystals. *Proceedings of the National Academy of Sciences*, 111(43), 15356–15361. <https://doi.org/10.1073/pnas.1411215111>] (Fig. 2), in which the mobility of the particles is explicitly tracked, to examine the contrast in mobility between different parts of the polycrystal.

Kind regards,

Dr Jack Panter

RESPONSE TO REVIEWERS' COMMENTS

Reviewer 1

Q1. *This is an experimental paper. The authors do not discuss their experimental approach neither in the abstract or in the introduction. The use of confocal microscopy and other techniques should be mentioned in the introduction. Is this a novel approach for defining the growth of defects during crystallization? The authors do not give enough background about the problem.*

A1. Thank you for the comment. This is indeed good to add for the broader audience as people outside the Soft Matter field will not know. We have included details about the experimental technique (optical/confocal microscopy) and references to its application to study crystal defects in the abstract and introduction as follows:

In the abstract: “We directly observe the formation and healing of common defects, like grain boundaries and vacancies **using confocal microscopy**.”

In the introduction: “One advantage of colloidal systems is that **defect formation [12-14] and dynamics [15, 16]** can be **conveniently** studied directly in real time with single particle resolution **using optical microscopy [17,18], ...**”

And at the end of the introduction: “We form **the graphene lattice** using pseudo-trivalent patchy particles} adsorbed at a substrate, and directly follow the crystallization, defect formation and healing with great temporal and spatial resolution **using confocal microscopy**”

Q2. *The authors state, “Our results reveal that the most prominent defect motif of colloidal and atomic graphene, a pentagon, is kinetically favoured in the early stages of graphene growth..”. I am not sure what the authors mean by kinetically favoured ? Do the authors think that pentagons are not equilibrium structures and thus they are kinetically trapped structures? If the attractive interaction is weak, then entropic forces might become considerable and the system might prefer the formation of pentagonal defects. Can they prove that the pentagons are not the result of minimization of the free energy at the weak interaction? Since the authors believe that pentagons are the kinetically trap structures, can they change the pathway?*

A2. The sentence “Our results reveal...” does indeed mean that we believe that pentagons are not equilibrium structures but kinetically trapped. In Supplementary note 6, we show the radial and bending potential between particles. Based on this, we expect the bending-energy penalty of 5-membered rings compared to 6-membered rings to be around $5k_B T$ at the conditions we are working. Assuming simple Boltzmann distribution, this means that there should be on the order of $1 / e^{-5} \approx 150$ times more 6-membered rings than 5-membered. This is clearly not the case.

Indeed, we have neglected entropic effects here, which will favour the 5-membered ring. We estimate the order of magnitude of these effects as follows: First, neglecting differences in the vibrational entropy of the 5 and 6 rings, the entropy gain of the 5-membered ring with respect to the 6-membered ring comes mainly from the presence of one unbound, free particle, giving rise to 3 additional degrees of freedom (2 translational, 1 rotational for the surface bound particles), contributing additional entropy of order $S \approx 3/2k_B$. If we take the vibrational entropy into account,

the difference will be smaller. Roughly, the number of normal modes of a two-dimensional N -particle ring is $2N - 3$, yielding 9 for a 6-ring, and 7 for a 5-ring. We then estimate that the total entropy difference, including vibrational effects, is of order $S \approx 1/2k_B$. The associated entropic part of the free energy $TS = 1/2k_B T$ is considerably smaller than the bending energy cost per ring, $U \approx 5k_B T$. At larger ΔT , the bending energy cost is lower and the difference between enthalpic and entropic terms smaller, but comparing the potentials at $\Delta T = 0.05^\circ\text{C}$ and $\Delta T = 0.2^\circ\text{C}$ in SI Fig. 7, we conclude that the change will not be too large. Experimentally, by systematic measurements of bending fluctuations of dipatch particles (Stuij *et al.*, *Soft Matter* **17**, 8291 (2021)), we determined an increase of the angular variance by a factor ~ 1.5 between $\Delta T = 0.05\text{K}$ and the lowest attraction, at which the bond eventually broke. The corresponding bending stiffness decreases by a factor $(1.5)^2 = 2.25$, meaning it would be $U \sim 2.2k_B T$ at the lowest possible attraction here, at which bonds are merely stable. This value is still larger than $TS = 0.5k_B T$. We thus conclude that it's mainly a kinetic effect that produces the 5-ring structures in the early stages, in agreement with 3D simulations on colloidal diamond.

We have indeed attempted to change the annealing pathway to exclude the formation of 5-membered rings, but not to great success. We have included an entirely new section in the SI (Sup Note 10), that discusses several strategies to circumvent the pentagon-problem. In addition, also in reply to the other reviewers, we have included in the main manuscript: “**To form a single crystal of atomic graphene, the single seed approach, nucleating and growing graphene from a single point, has been applied with success [52]. This single-seed approach is in principle applicable to our colloidal system owing to the temperature sensitivity of the critical Casimir interaction, yet may be difficult to achieve in practice due to the small temperature window. For colloidal systems, it has been suggested that the formation of pentagons could be suppressed by introducing two types A and B of patchy particles that only bind to the other type but not to themselves, forcing even-numbered rings [53]. While this is an elegant method, application to our critical Casimir system is not straight forward as interactions cannot easily be specifically encoded. We discuss these and more strategies and their suitability for the critical Casimir system in Supplementary Note 10.**”

Q3. How do the authors know that the attractive interaction between colloids is because of the Casimir forces? How about van der Waals forces? Can the authors explain how they calculate the magnitude of the Casimir forces in their system? Can the fluid deformation around the particles give a rise to an attractive force? Is there another system in which the casimir forces create a macroscopic lattice?

A3. Note that in this manuscript, we make use of the (admittedly confusing name) “*critical Casimir force*”, not the classic “Casimir force”. While the classical Casimir force is caused by quantum fluctuations, the *critical Casimir force* is the thermodynamic analogue, caused by compositional fluctuations of a binary solvent. The usage of the critical Casimir force for colloidal assembly is an established method in soft matter science to control colloidal interactions, see e.g. V. D. Nguyen et al. Critical Casimir forces for colloidal assembly, *J. Phys.: Condens. Matter* **28** 043001 (2016), and A. Maciolek, S. Dietrich, Collective behavior of colloids due to critical Casimir interactions, *Rev. Mod. Phys.* **90**, 045001 (2018). We have included these and further references in the manuscript to provide appropriate background by modifying the sentence as follows: “**The**

confinement of solvent fluctuations between the particle surfaces then causes attractive critical Casimir interactions on the order of the thermal energy, $k_B T$, tunable by the temperature offset ΔT from the solvent critical point, T_c [32-35].”

The pair potential between two particles consists of electrostatic repulsion due to charges on the particles, the critical Casimir force, and van der Waals forces; yet, the suspensions are always electrostatically stabilized against the latter (by design, so that permanent aggregation never occurs), and van der Waals forces are usually small at the particle distances considered. The critical Casimir force, however, varies with temperature, allowing us to reversibly control the colloidal interactions: At room temperature, the particles are stable by electrostatic repulsion, while close to the solvent critical point, the increasing solvent correlation length leads to increasing critical Casimir forces and particle aggregation. The total pair potential, as shown in SI Fig. 7a is obtained using critical Casimir scaling theory, fine-tuned to experimental results, see the original paper by Stuij *et al.*, *Soft Matter* **13**, 5233 (2017), and for the calibration onto our patchy particle system Jonas *et al.*, *J. Chem. Phys.* **155**, 034902 (2021). These references are included in the SI.

Q4. Have the authors tried different annealing pathways to overcome the energy barriers and to prevent the formation of defects when the attractive interaction is strong? If it is a kinetic trap, how universal their results are?

A4. Yes, we have repeatedly tried to obtain a less defected lattice using several annealing pathways, including simple heating-cooling cycles, very slow increase of attraction strength, and even attempts at introducing a temperature gradient (so the initial nucleation only occurs in a very limited region in the sample). Unfortunately, none of these methods seem to produce any significantly better results. We'd like to stress that indeed the strength of the current system precisely is the reversible temperature tunability of the potential, opening the door to these different annealing pathways. We have therefore added a new section to discuss these strategies in more detail - Supplementary note 10. See also our reply to question 2 and question 5 of reviewer 2.

*Q5. The authors mention, Simulations and experiment have shown that these colloidal molecules can grow into larger assemblies, yielding rich structures ranging from the kagome lattice to buckyball-like clusters. However, they refer neither to the experimental nor to the theoretical papers. For example, the paper by Weitz group: Grain Boundary Scars and Spherical Crystallography published in Science in 2003 discuss in detail why scars exist in spherical crystals. Or Li *et al.* analytical results show how disclinations interact with each other in curved space (PNAS 115, 10971–10976 (2018)) as in the case of buckyball and thus explain the symmetry of spherical viruses. In the curved systems, the presence of scars and pentagonal defects lowers the energy. In the system that the authors use, the minimum energy structure is defect-free. The paper will become more interesting if the authors compare the origin of defects in flat and curved space.*

A5. Thank you, we have added the suggested references. It is indeed interesting to compare assembly in our system with that of curved 2D systems. Colloids can crystallize into a hexagon on a flat surface, but not a curved one. However, introducing heptagons and pentagons to the lattice

makes it possible to tile a curved surface. On such surfaces, these “defects” are actually part of the minimum energy structure (R.E. Guerra et al. Nature 554, 346 (2018), A.R. Bausch et al. Science 299, 1716 (2003), W.T.M. Irvine et al. Nature 468, 947 (2010)). This is an interesting contrast to the penta- and heptagons we see in our 2D lattice, which are rather kinetically arrested, and not part of the equilibrium structure. We have added these thoughts at the end of the results section, just before the conclusions as follows: “It is interesting to compare the assembly in our system with that of curved 2D systems. Colloids can crystallize into hexagons on a flat surface, but not a curved one. However, introducing heptagons and pentagons to the lattice makes it possible to tile a curved surface (think of the pentagons on a football or on a viral shell) [54, 55]. On such surfaces, pentagons and heptagons are actually part of the minimum-energy structure [54, 56, 57]. This is an interesting contrast to our planar lattice, for which penta- and heptagons are defects, and not part of the equilibrium structure.”.

Q6. In supplementary Note 5, the authors indicate that the particle cluster grows linearly as a chain. Is this also a kinetic effect? This system should be completely out of equilibrium assuming a configuration with a considerable high line tension.

A6. Clusters grow linearly only in the very beginning (for very few number N of particles), where essentially only linear structures can form. According to Sup. Note 5, clusters grow mainly in one dimension for $N < 4$, transitioning to 2-dimensional growth when a cluster size of about 4 particles is reached. This can be explained by mere geometry of the bonds: clusters of up to 3 particles are always linear - branching is not possible, see Fig. A1 below (left). Adding a 4th particle, these clusters form either a linear chain, or a branched structure (or a ring, but very unlikely due to very high energy penalty). The branched structure is less likely to form than the linear structure, see Fig. A1 below (right): Starting from a linear triplet, there are 5 exposed patches. Only 1 of the 5 exposed patches leads to a branched cluster. Note that there is a significant presence of a smaller radius in Sup. Fig. 6b, consistent with this line of reasoning.

Fig. A1: Evolution of small clusters.

Adding a fourth particle to a three-particle linear cluster can result in 2 different structures: either a branched or a linear cluster. If the free particle bonds to any of the 4 patches pointed at in red, a linear cluster will form. If it bonds to the single patch indicated in blue, a branched cluster forms. Therefore, if each patch is equally likely to bond, we expect to observe the linear structures much more often.

For $N > 4$, the clusters form rings, and no longer prefer the linear structure, because clusters of size 5 or 6 almost perfectly align all patches for closing a loop.

Q7. The supplementary Note 6 is important for understanding the major energy components discussed in the main text. It is hard to understand the paper without the discussion of these energies in the main text. I recommend that the authors move some parts of Note 6 to the main texts. I think that the paper will be easier to follow if the authors showed the relevant equations and add some relevant theories to the paper.

A7. Thank you, we have moved supplementary figure S7a and b into the main text, together with part of the corresponding text.

Reviewer 2

Q1. The authors note that the pentagons and heptagons appear in pairs, as apparent in Figure 3. However, it appears in Figure 1g that the number of hexagons grows at the expense of pentagons as the strength of the attractions is increased, while the number of heptagons also shows an overall growth, albeit slightly. Can the authors comment on this observation?

A1. This is a good observation: the pentagons and heptagons appear in pairs in the solid phase, and this is why their numbers approach each other in Fig. 1h. However, initially in the fluid phase, single pentagons are the preferred motif, due to the kinetic effects described in the paper. Upon increasing attraction, bending energies become more pronounced and the number of pentagons and heptagons approach each other as they can form energetically more favorite units. Also, more hexagons form as the increasing attraction leads to increasing bending energy penalty of pentagons, thus increasing driving force to form hexagons. Hence, the number of pentagons decreases while the one of pentagon-heptagon combinations and hexagons increases, which are both energetically preferred. While the number of heptagons and pentagons approach each other, they do not quite reach each other. We associate that with pentagons lying primarily at the surface of grains; not all pentagons end up being paired with heptagons.

We have added a sentence about this in the revised manuscript as follows: “**This trend is reflected in the growth of heptagons in Figure 1h: pentagons can convert into hexagons, or promote the growth of neighbouring heptagons, which stabilizes the structure as a whole. This process is facilitated by the initial prevalence of pentagons.**”

Q2. The x-axis label of Figure 2b requires further clarity. It is somewhat different from what is stated in the figure caption and in the text. Perhaps a pictorial illustration will help.

A2. Thank you for noticing this inconsistency: We have renamed the x-axis to “Distance to interface/(#bonds)”, to make it consistent with the caption.

Q3. Given the importance of the analysis in terms of the bond bending energy, the authors should consider including a concise description of bond bending energy in the main body of the manuscript and state how it is calculated in the Methods section (rather than in the Supplementary

Material) so that the manuscript can be read seamlessly.

A3. Thank you. We have moved the bond bending SI Figures (7b and c) into the main text (see Figure 1d), together with explanatory text. We also added error bars. In the Supplementary note 6 and the Method section, we now go into more detail on how we construct the bending energy curve as well as show the histogram of observation, together with the requested statistical data.

Q4. In the concluding section, the authors argue that the energy scales are similar in atomic graphene and colloidal graphene in certain conditions. It is not clear then why the Stone-Wales defects, rather common in atomic graphene, are not seen in colloidal graphene. The comparison with atomic graphene with respect to the energy of formation of Stone-Wales defects should be elaborated.

A4. Thank you, we have elaborated on our analysis in the SI (Supplementary Note 7), where we treat the Stones-Waller defect. We hope that the revised text makes it clear why these defects, which are common in atomic graphene, are not expected in colloidal graphene, and indeed not observed.

Q5. A recent computational study [PNAS 118, e2109776118 (2021)] has highlighted the importance of the selection of six-member rings for the self-assembly of colloidal diamond. In this context, it will be worthwhile to comment on how the selection of six-member rings, in preference to five- and seven-member rings, along the crystallization pathways can possibly be encoded to improve the self-assembly kinetics as well as the quality of colloidal graphene thus formed.

A5. The cited work shows a very elegant approach to the problem, making non-6-membered rings impossible. This work has in fact inspired us to look for an experimental counterpart using our critical Casimir system; however, this approach is not easily realized in our system, due to the universal nature of the critical Casimir attraction: each patch will attract any other patch. Other techniques for assembly, specifically DNA-based attraction, could however offer suitable realizations to achieve these crystals via the proposed method. We believe that in our system, we need to resort to more “classical” optimization routes, such as using specific quench routes, the introduction of a single nucleation point, or using a template.

Based on this and other reviewer comments, we have added the reference and included a short discussion on how to improve lattice quality in the manuscript, and in more detail in Supplementary Note 10.

Specifically, in the manuscript we added: “To form a single crystal of atomic graphene, the single seed approach, nucleating and growing graphene from a single point, has been applied with success [52]. This single-seed approach is in principle applicable to our colloidal system owing to the temperature sensitivity of the critical Casimir interaction, yet may be difficult to achieve in practice due to the small temperature window. For colloidal systems, it has been suggested that the formation of pentagons could be suppressed by introducing two types A and B of patchy particles that only bind to the other type but not to themselves, forcing even-numbered rings [53]. While this is an elegant method, application to our critical Casimir system is not straight forward as interactions cannot easily be specifically encoded. We discuss these and more strategies and their

suitability for the critical Casimir system in Supplementary Note 10.”

Reviewer 3

Q1. ‘Colloidal graphene’ already has a different meaning in the graphene synthesis community, referring to a dispersion of graphene platelets [Li, L., & Yan, X. (2010). Colloidal Graphene Quantum Dots. *The Journal of Physical Chemistry Letters*, 1(17), 2572–2576. <https://doi.org/10.1021/jz100862f>]. Although this label makes sense in the main text, in the title it is misleading and could be passed over by the mesoscale assembly community.

A1. Thank you for noticing this possible confusion. We have replaced “assembling colloidal graphene” by “assembling the colloidal graphene lattice” in the title.

Q2. In Figs. 1e and 1f, the hexagonal labels (grey hexagons) are very hard to distinguish from the greyscale images – could they be recoloured?

A2. We have changed the colour of the hexagons to yellow throughout the manuscript.

Q3. Fig. 2c is referenced before Fig. 2b. Consider reorganising Fig 2.

A3. We have reorganized the figure to swap panels b and c in Fig. 2.

Q4. I find Fig. 2b and associated discussions confusing, and the explanation needs to be expanded. You say: To obtain insight into the bond bending strain and defect energy, we plot the bending energy as a function of distance away from the interface in Fig. 2b. Close to the grain boundary, there is an increased bending energy cost despite the energetic advantage of pentagon-heptagon combinations, maximizing at $0.5k_B T$ per particle. This energetic penalty quickly drops away from the interface, being virtually absent after just one layer of hexagons.

Plot the bending energy as a function of distance away from the interface – does this mean that: (1) you evaluate the average deviation in bond angles from 120° , and plot this against the distance away from the interface (2) you convert the bond angle deviation to bending energy, using supplementary Fig. 7b?

Furthermore, you say that you Plot the bending energy as a function of distance away from the interface, but Fig. 2b is a plot of the bending energy vs. ‘number of bonds removed from defect’. I don’t understand what you mean by this plot, or how it related to the discussion on bending energy vs distance from the interface.

How did you construct the bending energy as a function of distance away from the interface – did you observe a large number of interfaces and produce averages (in which case did you only select interfaces containing 5-7 defect chains, or were other types included in your sample)? Alternatively, is this evaluated from a single image? Either way, I would expect to see error bars in Fig. 2b representing the standard deviation about this average at each distance.

A4. To determine the bond bending energy in Fig. 2b, we use method (2) of the above, i.e. for each bond, we convert the bond angle deviation to bending energy and bin the data according to distance

to the interface to plot the resulting (average) bond bending energy as a function of distance to the interface. We apologize if the x-axis label has caused confusion; we have replaced it with “Distance to interface/(#bonds)” to be consistent with the caption. We have also clarified what we have plotted in the text, and added error bars. The revised text reads: “**To obtain insight into the bond bending strain and defect energy, we plot the bending energy as a function of distance away from the interface (in terms of number of bonds) in Fig. 2c. Close to the grain boundary, there is an increased bending energy cost despite the energetic advantage of pentagon-heptagon combinations, maximizing at $\sim 0.5k_B T$ per particle. This energetic penalty quickly drops away from the interface, being virtually absent after just one layer of hexagons.**”

Q5. Given the importance of the bending energy to the discussions in the main text, I find supplementary Figs. 7b and 7c highly important, and I feel they should be in the main text. The construction of supplementary Fig. 7b need to be explained more. Currently in the SI you say “By following three bonded particles and tracking the fluctuations of their bond angles, we can determine the bending energy by assuming a Boltzmann distribution. The resulting bending energy as a function of angle, determined from the probability distribution of bond angles, is shown in Supplementary Fig. 7b”. What was your sample size, and what are the uncertainties/error bars in Fig. 7b? Including more detail on this is important, e.g. showing the histogram and Boltzmann fit, as otherwise the reader cannot evaluate the reliability or accuracy of your bending energies.

A5. We have combined Figs 7b and c of the SI, added error bars, and moved it into the main text (see Figure 1d). In the method section, we briefly mention the process, and in Supplementary Note 6, we go into more depth on how we construct the bending energy curve as well as show the underlying histogram together with the requested statistical data, so that the reader can better evaluate the reliability of the data.

Q6. In Fig. 3d, the pentagon energy trace is red and the hexagon energy trace is blue, whereas throughout the rest of the paper, pentagons are identified as blue points, hexagons as grey, and heptagons as red. Can you make the colouring consistent?

A6. We thank the reviewer for pointing out this oversight. We have changed the colours throughout the manuscript to make them consistent, as suggested.

Q7. Fig. 4 doesn't appear to support your distinction between 'static and dynamic' grain boundaries. You claim that, The red and yellow delineated regions show examples of static and highly dynamic grain boundaries, respectively. The former shows no reorganization: any translation or rotation matches the movement of the entire crystal and the grain boundary is completely frozen. In contrast, the yellow delineated region close to the junction of multiple grains shows significant reconfiguration.

However, observing Figs. 4d-f, the rearrangement dynamics appear quite similar, (particularly Fig. 4f). To investigate your claim, I would expect to see an analysis similar to, for example, [Singh, N., Sood, A. K., & Ganapathy, R. (2020). Cooperatively rearranging regions change shape near the mode-coupling crossover for colloidal liquids on a sphere. Nature Communications,

11(1), 4967. <https://doi.org/10.1038/s41467-020-18760-7>] (Fig.3),

or

[van der Meer, B., Qi, W., Fokkink, R. G., van der Gucht, J., Dijkstra, M., & Sprakel, J. (2014). Highly cooperative stress relaxation in two-dimensional soft colloidal crystals. *Proceedings of the National Academy of Sciences*, 111(43), 15356–15361. <https://doi.org/10.1073/pnas.1411215111>] (Fig. 2),

in which the mobility of the particles is explicitly tracked, to examine the contrast in mobility between different parts of the polycrystal.

A7. To better quantify the single-particle dynamics and its difference between the ‘static and dynamic’ grain boundaries, we have performed some additional analysis, resulting in the new Supplementary Note 11, as described below.

The specific measures of dynamics suggested by the reviewer above rely on accurate linking of particles (i.e. particle A is always identified as being particle A), which unfortunately is challenging in this very long (9 hour) experiment recorded at a relatively low framerate, to prevent photobleaching of the fluorescent dye: While we can track particles in the static lattice easily, we quickly lose track of non-bonded, dynamic particles, making this direct particle tracking prohibitively difficult.

We therefore reverted back to the raw images to obtain insights into the different dynamics of the lattices: We first stabilize the view on the lattice by removing drift and other collective movement from the data (this is relatively straightforward using the ImageJ StackReg plugin). Then, for all images, we calculate the standard deviation and mean intensity of each spot/pixel in the field of view. The ratio between the standard deviation σ and mean intensity $\langle I \rangle$, $K = \sigma / \langle I \rangle$ can then be used to quantify the dynamics at every spot in the sample. The contrast image is shown in Supplementary Fig. 10. In panel a on the right, we show the contrast image of the red region of Fig. 4 of the main text, while in panel b on the right, we show that of the yellow region. In these images, low values (blue) indicate little change in intensity, meaning a static structure, while high values (yellow) indicate strong intensity fluctuations, and thus a dynamics structure. Apparently, the static (red) region shows indeed little mobility over the 9-hour measurement interval. In contrast, the dynamic (yellow) region shows some distinct high-mobility regions indicated by green to yellow colours, where particles have apparently spent time bonding and de-bonding. Some longer-lived (but still transient) structures can be distinguished in the central region from a darker colour (dotted red line guiding the eye).

Performing this analysis on the entire polycrystal of Fig. 4 is unfortunately not straightforward due to the non-uniformity of the drift. While drift movement can be accounted for in small regions with matching translational and rotational drift, in the full poly-lattice, individual crystals drift relative to each other on experimental timescales, making this analysis difficult.

We hope this additional analysis shows the difference in rearrangement dynamics more clearly to a satisfying degree. We have included the above discussion in the SI (Supplementary Note 11), and added a sentence linking to it in the main text as follows: “**The enhanced dynamics is confirmed in a more detailed particle-scale analysis as shown in Supplementary note 11.**”.

REVIEWERS' COMMENTS

Reviewer #1 (Remarks to the Author):

The authors have answered to all my questions and have addressed all my concerns. Thus I recommend it for publication in the current format.

Reviewer #2 (Remarks to the Author):

The authors have satisfactorily revised the manuscript by including additional analyses and discussions and providing further clarifications in response to the Reviewers' comments. The revised version, as it stands, presents an important piece of work, which will appeal to a broad readership that Nature Communications enjoys. Hence, I recommend publication of this manuscript in Nature Communications in its present form.

Reviewer #3 (Remarks to the Author):

I thank the authors for addressing each of my points so thoroughly. The addition of Supplementary Note 11 and Supplementary Figure 10 in particular has strengthened the discussion of the dynamics of defects at the grain boundaries.

I would now like to recommend this interesting manuscript for publication.